# Global Transcriptional Response of Three Highly Acid-Tolerant Field Strains of *Listeria monocytogenes* to HCl Stress

**DOI:** 10.3390/microorganisms7100455

**Published:** 2019-10-16

**Authors:** Jule Anna Horlbog, Marc J. A. Stevens, Roger Stephan, Claudia Guldimann

**Affiliations:** Institute for Food Safety and Hygiene, Vetsuisse Faculty, University of Zurich, 8006 Zürich, Switzerland; jule.horlbog@uzh.ch (J.A.H.); marc.stevens@uzh.ch (M.J.A.S.); stephanr@fsafety.uzh.ch (R.S.)

**Keywords:** *Listeria monocytogenes*, acid stress, HCl, tolerance, RNAseq, cytochrome C, catabolite repression, biofilm, flagella

## Abstract

Tolerance to acid is of dual importance for the food-borne pathogen *Listeria monocytogenes*: acids are used as a preservative, and gastric acid is one of the first defenses within the host. There are considerable differences in the acid tolerance of strains. Here we present the transcriptomic response of acid-tolerant field strains of *L. monocytogenes* to HCl at pH 3.0. RNAseq revealed significant differential expression of genes involved in phosphotransferase systems, oxidative phosphorylation, cell morphology, motility, and biofilm formation. Genes in the acetoin biosynthesis pathway were upregulated, suggesting that *L. monocytogenes* shifts to metabolizing pyruvate to acetoin under organic acid stress. We also identified the formation of cell aggregates in microcolonies as a potential relief strategy. A motif search within the first 150 bp upstream of differentially expressed genes identified a novel potential regulatory sequence that may have a function in the regulation of virulence gene expression. Our data support a model where an excess of intracellular H+ ions is counteracted by pumping H+ out of the cytosol via cytochrome C under reduced activity of the ATP synthase. The observed morphological changes suggest that acid stress may cause cells to aggregate in biofilm microcolonies to create a more favorable microenvironment. Additionally, HCl stress in the host stomach may serve as (i) a signal to downregulate highly immunogenic flagella, and (ii) as an indicator for the imminent contact with host cells which triggers early stage virulence genes.

## 1. Introduction

Most human cases of listeriosis are caused by the ingestion food contaminated with *Listeria monocytogenes*. The disease often has a mild course in healthy individuals, while the elderly, young children, pregnant women and immunocompromised people may suffer life-threatening invasive forms of listeriosis like sepsis, meningitis, or abortions [1].

During the transition along the food production chain and into the human host, *L. monocytogenes* is able to survive food safety measures and host defense mechanisms. Low pH is of special interest in this context because of its dual role as a food preservative and as a first line of defense in the gastrointestinal tract of the human host.

*L. monocytogenes*, a low-CG Gram positive organism, has a remarkable tolerance against low pH conditions [2]. This is especially relevant in food in which acid serves as a preservative, but also because strains with a higher acid tolerance might survive the low pH in the human stomach at an increased rate and therefore pose a higher risk for infection. Therefore, the potential hazard associated with a given strain of *L. monocytogenes* found in the food environment is affected by its degree of acid tolerance.

In general, *L. monocytogenes* cells faced with the challenge of an increased concentration of H+ ions have the options of (i) consuming H+ ions, e.g., through decarboxylation reactions, (ii) exporting H+ ions through pumps, and (iii) repairing the damage resulting from low pH, for instance mediated by heat shock proteins. All these mechanisms have been described for *L. monocytogenes* [2,3], making the acid response of *L. monocytogenes* a complex, highly orchestrated process that involves various components. Many genes involved in stress response in *L. monocytogenes* are regulated by the alternative sigma factor B (SigB) [4] and indeed *sigB* mutants are highly acid sensitive [5,6,7]. The glutamate amino acid decarboxylase Gad is an example for SigB-regulated acid response [8,9,10]. Gad catalyzes the formation of γ-aminobutyric acid (GABA) from glutamate in a reaction that consumes protons [11]. Other known responses to acid shock in *L. monocytogenes* include the SOS response regulated by RecA [12,13], the LisRK two-component system [14], the agmatine deiminase system [15], proton consumption by acetoin production [2], and, in exponential growing cells, using the F_0_/F_1_-ATP synthase in reverse to pump protons from the cytosol to the extracellular space under the expense of ATP [16].

Most work on acid tolerance in *L. monocytogenes* was performed using laboratory strains that have been passaged in research institutions for years [17], and few studies exist that compare the acid tolerance between different strains [18,19,20,21]. We have previously shown large differences in acid tolerance between field strains of *L. monocytogenes* isolated from the food and host environment [22]. Interestingly, the acid phenotype was largely strain specific and not consistently associated with lineage, serotype or clonal complex. Here, we set out to determine the mechanism by which three highly acid-tolerant field strains achieve survival under acid shock at pH 3.0 with HCL, mimicking the acidic conditions encountered during the transition from the food environment to infection of the human host. The global transcriptional response under acid stress was compared to non-stress conditions. The strains used in this study were isolated from patients or food products and are thus directly related to real-life settings.

## 2. Results and Discussion

### 2.1. General Response of 3 Strains to Acid Stress

*L. monocytogenes* in early stationary phase was exposed to non-lethal acid stress with HCl at pH 3.0 for one hour, as previously described [22], and differentially expressed genes were determined in comparison to non-acid exposed controls. While the three acid-resistant strains used here ceased growing under these conditions, a large proportion of cells survived (−0.5–1 log CFU/mL, compared to −4–7 log CFU/mL in acid-sensitive strains) [22]. The number of genes that were significantly up- or downregulated were similar for LL195 (394/347 up and down regulated, respectively) and N12-0605 (380/227), but higher for N13-1507 (764/689). The core acid regulon of differentially expressed genes consisted of 150 upregulated genes and 169 downregulated genes (Figure 1, Appendix A). From these gene counts, two major conclusions were drawn: (i) A considerable proportion of 319 differentially expressed genes was shared across all strains, and (ii) the acid regulon of strain N13-1507 was larger than that of LL195 and N12-0605. This difference in regulon size is likely an effect of the different genetic background of strain N13-1507 (CC6) compared to LL195 and N12-0605 (CC1). An overview of gene ontology terms associated with the up- and downregulated genes can be found in Appendix A. Our further analysis focused primarily on differentially expressed genes that were shared across at least two strains in order to characterize the core acid regulon under HCl stress.

### 2.2. Components of the Acid Response in L. monocytogenes were Differentially Expressed after Acid Stress

A number of the genes that were differentially expressed after acid stress correspond to genes previously described as involved in the acid response of *L. monocytogenes*, namely the activation of the general stress response transcriptional regulator SigB, the Gad system, parts of the SOS response and the agmatine deiminase system [5,6,7,8,9,10,12,13,15].

The activation of SigB was evident in the increased transcription of *lmo2230* (*L. monocytogenes* EGDe nomenclature) after acid stress in strains LL195 and N13-150. *lmo2230* encodes an arsenate reductase that is under direct SigB control and has previously been used as a reporter for SigB activity [23,24,25].

Another SigB-regulated gene whose product is involved in the acid response of *L. monocytogenes* is the glutamate amino acid decarboxylase *gad* [26]. Amino acid decarboxylases consume protons and their activation helps alleviate intracellular acid stress. Transcripts for *gad* were increased in LL195 and N13-1507 after acid exposure. Other amino acid decarboxylases like the glycine decarboxylases encoded by *gcvPB* and *gcvPA* were upregulated in all three strains, and the arginine decarboxylase encoded by *yaaO* was upregulated in LL195 and N13-1507.

The SOS response is an inducible pathway involved in DNA repair [27], and *L. monocytogenes* with a deficient SOS response system (*recA* mutants) show impaired survival under acid stress [13]. The gene coding for the main transcriptional activator of the SOS response, *recA* [13], was not induced by acid stress. However, *yneA* is part of the *L. monocytogenes* SOS response [13] and was upregulated in all three strains.

In addition to the above-mentioned mechanisms, *L. monocytogenes* have been shown to use the agmatine deiminase system to alleviate acid stress [28]. Agmatine catabolism produces ATP and releases NH3, which helps raise the intracellular pH. In our dataset, the agmatine deiminase encoded by *aguA1* was upregulated in N13-1507 after acid stress.

### 2.3. Genes Involved in Cell Division and Gene Expression were Downregulated after Acid Stress

After comparing the transcriptional response to known acid responses, we used KEGG [29], gene ontology (GO) enrichment [30] and manual analysis to identify novel components of the acid response in these three *L. monocytogenes* field strains (Appendix A).

Genes whose products are involved in genome replication and partition (*dnaG*, *smc*) and genes coding for cell envelope components (*fabG*, *dat*, *tarL*, *tagB*, *tagF*, and *dltCD*) were downregulated in all three strains in the post-acid dataset, which is in agreement with the observed growth arrest. GO term analysis showed that the GO terms “gene expression” (GO:0010467), “ribonucleoside triphosphate biosynthetic process” (GO:0009201), several terms associated with tRNA aminoacylation (GO:0006399, GO:0006418, GO:0043039), “aminoacyl-tRNA ligase activity” (GO:0004812), “translation” (GO:0006412), and several terms associated with ribosomal components (GO:0003735, GO:0015934, GO:0015935) were enriched among the downregulated genes in all three strains, which again corresponds with the growth arrest and points toward an overall decreased metabolic activity of the cells. For instance, transcripts for the *rpoA* RNA polymerase subunit were downregulated in all three strains post acid stress. Reduced translation was apparent in the downregulation of 26 ribosomal proteins, 14 tRNA ligases, the translation initiation factors *infA* and *infC*, and the elongation factor *fusA* in all three strains.

### 2.4. Changes in Metabolism after Acid Stress

All strains showed profound changes in carbon metabolism after acid stress. Among the top upregulated genes, a cluster of phosphotransferase systems (PTSs) was evident. A Fisher’s exact test confirmed that PTS were significantly upregulated in all three strains (*p* < 0.0001 in LL195 and N13-1507, *p* = 0.028 in N12-0605). Specifically, the EIIA components of fructose, mannose, galacitol, lactose, and glycerol PTSs were upregulated after acid stress in all strains. Additionally, the transcriptional regulator of a cellobiose PTS, *licR*, and all components of the glycerol catabolism pathway (*glpF*, *golD*, *dhaKLM*) were upregulated in all three strains in the post-acid dataset.

Upregulation of PTS systems and the glycerol catabolism pathway has been observed in other organisms, for instance in the low-G+C Gram positive *Lactococcus lactis*, where exposure to pH 5.1 resulted in a significant upregulation of a cellobiose PTS [31]. This may indicate an increased demand of alternative carbon sources by the cells. The experiments were performed in BHI with an initial glucose content of 2 g/L and glucose starvation is an unlikely cause for this apparent need for alternative carbon sources. The upregulation of PTS under acid stress therefore indicates a release of the carbon catabolite repression (CCR) under acid stress. CCR is mediated by a global transcriptional regulator: The catabolite control protein A (CcpA), which is activated by HPr phosphorylated at serine 46 (HPr-ser46). Phosphorylation of HPr is catalyzed by the HPr kinase/phosphatase (HprK/P), which is a sensor of the energy state within the cell (reviewed in [32]). A study in *B. subtilis* [33] correlated HprK/P activity with pH and showed that lowering the reaction pH lead to a shift towards phosphatase activity in HprK/P. As a consequence, HPr-ser46 was not phosphorylated, and hence CcpA did no longer repress the catabolite promoters. Because CCR is highly conserved in firmicutes [34] a similar mechanism may be responsible for the observed de-repression of the CCR in *L. monocytogenes*.

In addition, the fumarate reductase flavoprotein subunit, which is part of the citric acid cycle, was downregulated after acid stress in all strains.

### 2.5. Shift in Pyruvate Fermentation

*L. monocytogenes* can ferment pyruvate to lactate in the presence of abundant NADH. In the absence of NADH, fermentation shifts to the reaction from pyruvate to acetoin via 2-acetolactate in reactions that use one proton each [35]. The reaction from pyruvate to 2-acetolactate is catalyzed by the acetolactate synthase encoded by *ilvB*, *ilvH*, and *alsS*, the reaction from 2-acetolactate to R-acetoin by alpha-acetolactate decarboxylase encoded by *alsD.* Of these, *alsD* was upregulated after acid stress in all three strains. *alsS* was upregulated in N13-1507 and LL195, and *ilvB* was upregulated in N13-1507. This shift to producing acetoin may alleviate acid stress by consuming protons and by not using the alternative pathway of fermenting pyruvate to lactic acid [36,37]. *AlsS* and *alsD* were also upregulated after exposure of *L. monocytogenes* EGDe [38] and H7858 [39] to organic acids.

### 2.6. Genes Involved in Cell Wall Biology and Biofilm Formation were Differentially Expressed after Acid Stress

The GO terms “membrane” (GO:0016020), “cell wall” (GO:0005618) and “plasma membrane” (GO:0005886) were enriched among the genes that were downregulated after acid stress in all three strains, consistent with growth arrest. In contrast, more than ten GO terms associated with the cell envelope (i.e., GO:0005887, GO:0016021, GO:0020002, GO:0031224, GO:0031226, GO:0032977, GO:0033644, GO:0044218, GO:0044279, GO:0044425 GO:0051205, GO:0051668, GO:0061024, GO:0016998) were enriched in the set of upregulated genes after acid stress in at least one strain. This suggests profound remodeling of the cell envelope in response to acid stress. In addition, several genes coding for proteins involved in cell wall functions, such as the SOS response member *yneA*, the cell shape factor *ylmH* and the membrane insertion factor *yidD* were upregulated in all strains, while the membrane insertion factor *yidC* was upregulated in N13-1507.

Comparative microscopy of Gram-stained *L. monocytogenes* cells before and after acid stress suggested that *L. monocytogenes* elongated and aggregated with acid stress (Appendix A), and an image stream analysis confirmed that individual cells were larger after acid stress (*p* < 0.001) (Figure 2a).

Cell elongation under different stress conditions has been shown for a variety of organisms and stress conditions; e.g., in *Salmonella* Enteritidis after cold stress [40], in *L. monocytogenes* after salt stress [41] and in *L. monocytogenes*, *Bacillus* and *Clostridium* species after acid stress [42]. In *L. monocytogenes* and *B. subtilis*, induction of *yneA* by the SOS response resulted in elongated cells [13,43], possibly by the accumulation of YneA at the cell center which prevented septum formation [43]. YlmH has been described in *B. subtilis* as a “factor involved in shape determination” [44,45] and the *ylm* cluster of genes has been speculated to be involved in cell division [46]. Hence it is conceivable that the combined effect of increased transcription of *ylmH* and *yneA* may have caused the observed cell elongation after acid stress.

The image stream analysis also revealed an increase in aggregates of cells in the post acid stress samples compared to the controls (Figure 2b). *LuxS* and *yneA* were upregulated under acid stress in all three strains. The products of these genes are involved in early and late stage biofilm formation, respectively [47,48,49]. *AgrD*, another early stage biofilm gene [50], was upregulated in N12-0605 and N13-1507 after acid exposure. Biofilm formation and secretion of extracellular matrix may offer an explanation for the observed increase of cell clumping after acid exposure, since the matrix may cause cells to stick together in biofilm microcolonies.

Remarkably, the biofilm dispersion locus A (*bldA*) (responsible for dispersion of cells from mature biofilm by lowering their adherence in *Pseudomonas aeruginosa* [51]) was downregulated in LL195 and N13-1507 after acid stress.

Taken together, our data support a model where the formation of biofilm microcolonies is used as a strategy to combat acid stress by creating a slightly more favorable microenvironment in the immediate vicinity of cells. *L. monocytogenes* showed more efficient adherence to stainless steel when grown in medium adjusted to pH 6.0 compared to pH 7.3 [52], while pre-exposure to acidic conditions did not affect the biofilm formation ability at neutral pH [53].

Our study describes, to the best of our knowledge, for the first time the formation of aggregates of *L. monocytogenes* stationary phase cells after acid stress.

### 2.7. Genes Involved in Cellular Respiration were Differentially Expressed after Acid Stress

Expulsion of protons from the cytoplasm is an effective strategy to relieve acid stress [2]. During aerobic respiration, the electron transport chain mediates the translocation of protons across the cell wall. This generates a proton gradient that is subsequently used to drive the ATP synthase. GO term enrichment analysis showed that several components of this system were overrepresented among the upregulated genes (“electron transport chain” GO:0022900, “respiratory electron transport chain” GO:0022904, “ATP synthesis coupled electron transport” GO:0042773). The cytochrome BC complex uses redox reactions between quinone/quinol to power the electron transport chain, with molybdopterins acting as cofactors in the quinone/quinol cycling [54,55]. Genes coding for quinol oxidase subunits three and four (*qoxC* and *qoxD*) were upregulated in all three strains, while genes coding for quinol oxidase subunits one and two (*qoxB* and *qoxA*) were upregulated in LL195 and N13-1507. Molybdopterin *moaB* was upregulated in all three strains after acid exposure, while *moeA*, *mobB*, *moaEI* and *moaD* were upregulated in LL195 and N13-1507. Cytochromes depend on insertases for their correct insertion into cell membranes [56]. In line with this, *yidD* has been upregulated in all three strains after acid stress. *yidD* codes for a membrane protein insertion efficiency factor, and YidD in concert with YidC have been shown to be involved in the insertion of cytochrome oxidases as well as the F_0_c subunit of the ATPase into the membrane in *Escherichia coli* [57].

These results suggest that *L. monocytogenes* is using the electron transport chain to pump protons out of the cytosol to relieve acid stress. An analogous mechanism has been speculated for *Bacillus cereus* and *Bacillus subtilis* under acid stress [37,58]. In contrast, *L. monocytogenes* exponential phase cells, but not stationary phase cells, use the ATP synthase in reverse to pump protons out of the cytosol under the use of ATP [16].

The proton gradient created through respiration is used by the ATP synthase to drive ATP synthesis. GO term analysis showed that genes encoding subunits of the ATP synthase were downregulated after acid stress (“proton-transporting ATP synthase complex” GO:0045259, “plasma membrane proton-transporting ATP synthase complex” GO:0045260, “proton-transporting ATP synthase complex, catalytic core F(1)” GO:0045261, “plasma membrane proton-transporting ATP synthase complex, catalytic core F(1)” GO:0045262). Specifically, *atpC*, *atpH*, *atpD2*, *atpG*, and *yscN* were downregulated in all three strains after acid stress, and all additional genes that belong to the ATP synthase complex (*atpC*, *atpA2*, *atpE*, *atpC*, *atpI*, *atpF*) were downregulated in at least one strain after acid stress. The efficiency of the respiratory chain to pump protons out of the cytosol, as proposed above, would to a degree be abolished if the ATP synthase is then used to import H^+^ ions into the cell again. The observed downregulation of transcripts coding for the ATP synthase can therefore be seen as a strategy to increase the efficiency of stress relieve generated by the upregulation of proton pumps in the respiratory chain. Interestingly, high pH stress in *E. coli* results in F_1_F_0_ ATP synthase upregulation, suggesting that the ATP synthase was used to import H^+^ [59]. Arguably, downregulating the ATP synthase would result in decreased ATP production solely via glycolysis. However, we found no evidence of upregulation of alternative pathways for ATP synthesis. This may explain the observed growth arrest and reduced metabolism.

Also, the increased activity of the respiratory chain regenerates more NAD from NADH. Hence, NAD regeneration reactions such as lactate formation from pyruvate are not needed. This enables the observed activation of the proton consuming pathway from pyruvate to acetoin.

Taken together, these results show for the first time that *L. monocytogenes* stationary phase cells under HCl stress increase the expression of proton pumps in the respiratory chain, while at the same time reducing the activity of the ATP synthase, resulting in a net loss of H^+^ from the cells.

### 2.8. Cell Motility Related Genes were Downregulated after Acid Stress

All three strains used in this study were motile at 37 °C (Appendix A). The GO term “cell motility” (GO:0048870) and eight GO terms associated with different parts of the flagella (GO:0001539, GO:0009288, GO:0009420, GO:0009424, GO:0009425, GO:0030694, GO:0044461, GO:0055040) were enriched among genes that were downregulated after acid stress. Accordingly, genes involved in cellular motility, e.g., flagellar synthesis genes (*flaA*, *fliEFGM*, *flgB*, *flgC*, *flgEGKL*, *motB* and *lmo0707*) and genes involved in chemotaxis (*cheARVY*, and *pomA*) were downregulated in all three strains after acid stress.

Repression of flagella is in line with the previously described repression of *flaA* or the gene sets “cellular processes: chemotaxis and motility” repressed after long term exposure to pH 5.5 [60,61].

Flagella are strong immunogens [62] and consume considerable amounts of energy [63]—hence it makes evolutionary sense for a pathogen to tightly control their expression. Motile *L. monocytogenes* 10403S outcompeted otherwise isogenic, non-motile Δ*flaA* mutants in vivo [64], and exposure to bile in the duodenum induced the expression of motility genes in *L. monocytogenes* 10403S and H7858 [65].

Combined with these other studies, our data support the hypothesis that flagella are suppressed during exposure to acid mimicking the first stages of infection, i.e., the exposure to HCl in the stomach (this study). Upon entry into the duodenum, flagella are expressed to facilitate the physical contact with host cells [64], possibly by using bile as a signal for imminent host cell contact [65].

### 2.9. Early-Stage Virulence Genes were Upregulated after Acid Stress

In a next step, the response of virulence genes to acid was studied. Transcripts for the positive regulatory factor A (*prfA*) were upregulated in post-acid stress for N13-1507 and LL195. PrfA is the main transcriptional regulator of virulence genes in *L. monocytogenes* [66]. Further, genes that belong to the PrfA regulon, *inlA* and *hly*, were highly induced in all three strains after acid stress. *hly* codes for the pore-forming toxin listeriolysin O [67], while *inlA* encodes an internalin that is important for cell adhesion and host cell invasion [68]. Both the products of *inlA* and *hly* are involved in early steps of host cell invasion. In contrast, virulence genes involved in later steps of host cell invasion such as *plcA*, *actA*, *mpl*, and *plcB* were not induced by acid stress. Additionally, in order to be in its active form, PrfA needs glutathione as a cofactor, and both PrfA itself and glutathione need to be reduced by thioredoxins and glutathione reductases [69,70]. Remarkably, three thioredoxins (*trxA*, *trx1* and *lmo2152*) and *trxB* (thioredoxin reductase) were upregulated in all three strains. Taken together, our data indicate that the exposure of *L. monocytogenes* to HCl during the gastric passage could lead to an induction of parts of the PrfA regulon, specifically InlA which is used to attach to host cells and LLO which is needed to escape the phagocytic vacuole. PrfA itself is regulated by a positive feedback loop and it is positively co-regulated by SigB [71]. The increased amounts of active SigB as well as activated PrfA after acid stress may jointly have led to increased *prfA* transcription, which is also evident in other studies that found *prfA* itself or PrfA-dependent genes upregulated after exposure of *L. monocytogenes* to organic and inorganic acids [39,60,61,72,73].

### 2.10. Promoter Analysis of Differentially Expressed Genes

A motif search within the first 150 bp upstream of differentially expressed genes identified the involvement of two potential regulatory sequences in the regulation of the acid stress response (Figure 3a). The genome sequences of all strains were then screened for these motifs to identify potential co-regulation of genes by novel regulatory mechanisms.

Motif 14_0047 is a palindromic sequence that is homologous to the consensus binding sequence for the transcriptional regulator of ferric uptake Fur in *B. subtilis* [74]. It was identified in six instances in LL195 and five in N12-0605 and N13-1507, respectively (Table 1). Consistent with the homology to the *B. subtilis* Fur box, motif 14_0047 was found in front of several genes that are annotated with functions related to iron transport and management. In LL195, these were genes coding for the Iron(3+)-hydroxamate-binding protein *yxeB*, the iron transporter *feoB*, and the Fe-S protein maturation auxiliary factor *sufT*, of which *feoB* was upregulated after acid stress. Additionally, motif 14_0047 was found in front of an O-acetyl-ADP ribose homologous to *lmo2759*, which was also upregulated in the post-acid dataset. In N12-0605 and N13-1507, motif 14_0047 was found in front of the Fe(3+)-citrate-binding protein *yfmC* and *sufT*, both of which were upregulated after acid stress. Interestingly, upregulation of the iron transporter FeoB was also observed after exposure of *L. monocytogenes* to organic acids [39], and *Corynebacterium glutamicum* induced the iron-starvation response at pH 6.0 [75] and pH 5.7 [76] compared to neutral conditions.

Complex media such as tryptic soy broth and BHI sequester iron at low pH [76]. Therefore, the observed induction of iron transporters is likely not a direct effect of the acid response, but rather of the iron starvation induced by performing acid stress at low pH in BHI.

Motif 17_1507 was found in 15 instances in all three strains (Table 1). Notably, it was identified upstream of *plcA* and thioredoxins (*lmo2152*). In the case of *plcA*, the motif included the PrfA-box upstream of *plcA* (Figure 3b). Motif 17_1507 was also identified in front of glutathione reductases (*lmo1433)*, which are involved in the mitigation of oxidative stress [77]. The presence of this newly identified motif 17_1507 in the promoter regions of thioredoxins and *plcA* is consistent with a role in PrfA regulation. Motif 17_1507 upstream of *plcA* included the PrfA box, which is responsible for positive autoregulation of PrfA via transcription of a bicistronic mRNA containing *plcA* and *prfA* [78]. Loss of the readthrough from the PrfA-box upstream of *plcA* leads to loss of virulence [79]. Whether this motif has a regulatory function and whether the binding of a transcriptional regulator to this motif has a positive or negative effect on transcription of downstream genes remains to be determined.

In conclusion, RNAseq of HCl-exposed, early stationary phase *L. monocytogenes* cells shows a broad reshaping of the global transcriptional profile in response to acid shock with HCl at pH 3. Compared to the controls, many downregulated genes suggested growth arrest, slower metabolism and the reduction of motility in the post-acid dataset.

Our data support a model where *L. monocytogenes* combats the influx of H^+^ by using proton pumps and proton-consuming processes such as amino acid decarboxylases, and by forming biofilm microcolonies that could serve to establish a protective microclimate. During host invasion, the low pH in the stomach may serve as a signal to induce the staggered expression of virulence genes, possibly via activation of PrfA and a potential regulatory effect of the newly identified motif upstream of virulence-associated genes.

## 3. Materials and Methods

### 3.1. Bacterial Strains

Three lineage I, serotype 4b, acid-resistant strains from Horlbog et al. [22] were included in this study (Table 2). LL195 is a well characterized strain from an outbreak in Switzerland [80]. We included a closely related CC1 strain isolated from a meat product in 2012 (N12-0605) and a CC6 strain isolated from a human patient in 2013 (N13-1507). Both N12-0605 and N13-1507 were obtained from the Swiss National Reference Centre for Enteropathogenic Bacteria and Listeria (NENT).

### 3.2. Growth Conditions

*L. monocytogenes* were maintained as 15% glycerol stocks (Sigma-Aldrich, St. Louis, Missouri, United States) at −80 °C. To obtain cultures, they were streaked onto BHI agar plates (Oxoid, Hampshire, UK) and incubated at 37 °C for 16–18 h. A single colony was inoculated into 5 mL BHI and incubated at 37 °C for 16–18 h with shaking, sub cultured 1:100 into 5 mL of fresh BHI and grown for 6 h at 37 °C with shaking to obtain an early stationary culture corresponding to OD_590_ 1.0 plus one hour. Acid stress was performed by adding 5 mL BHI adjusted to pH 1.6 with HCl (Fluka, Bucharest, Romania). This ratio had been previously titrated to result in a final pH of 3. Our previous study showed that pH 3 is able to differentiate between acid resistant and acid sensitive strains [22]. Controls were diluted 1:1 with untreated BHI. During the acid stress, all tubes were incubated at 37 °C for one hour without shaking. To avoid artefacts due to variations in the BHI composition, all liquid BHI used in this study originated from the same batch of pre-prepared BHI.

### 3.3. RNA Isolation

Strains LL195 and N12-0605 were sampled in biological triplicates whereas N13-1507 was represented in duplicate due to flow cell and sequencing depth limitations. After one hour of acid exposure, or the respective control condition in BHI without acid, all reactions were stopped by the addition of 1 mL of 10% acid phenol-chlorophorm in ethanol (*v*/*v*) to the cultures, which were chilled to 4 °C immediately. To extract the total RNA, the cultures were pelleted at 20,000× *g* at 4 °C, resuspended in 200 µL of 10 µM Tris-buffer (Sigma-Aldrich, Buchs, Switzerland) containing 20 mg/mL lysozyme (Sigma-Aldrich) and 50 µL of 20 mg/mL proteinase K (Qiagen, Hilden, Germany) and subsequently lysed for 30 min at 37 °C. Cell lysates were then mixed with 1 mL TRI-Reagent (Invitrogen by Thermo Fischer Scientific, Carlsbad, California, United States) and bead-beaten for 2 × 60 s with cooling on ice in between in beat-beating tubes containing 1.4 mm ceramic beads (MagNA lyser green tubes, Roche, Basel, Switzerland). Subsequently the samples were centrifuged at 16,000 × *g* at 4 °C and the supernatant was transferred to a new tube, where 500 µL of bromo-chloro-propane (BCP, ACROS Organics, Geel, Belgium) were added. After 10 min incubation at room temperature the tubes were centrifuged for 15 min at 16,000 × *g* at 4 °C and the top aqueous layer was transferred into a new tube. Nucleic acids were precipitated overnight in 2.5 mL isopropanol at −80 °C. The following day, nucleic acids were pelleted, washed with 5 mL 75% ethanol, resuspended in nuclease free water and adjusted to 500 ng/µL using a nanodrop (Witec AG, Sursee, Switzerland). 3 µL Turbo DNAse (Invitrogen by Thermo Fischer Scientific) and its buffer were added to digest the genomic DNA for one hour at 37 °C, with the addition of another 3 µL Turbo DNAse after 30 min to ensure complete digestion of genomic DNA. The DNAse treatment was followed by a phenol-chloroform (Fluka) extraction and RNA precipitation in EtOH with 1.6% (*v*/*v*) 3M sodium acetate at −80 °C overnight. The RNA was pelleted and washed with 70% ethanol, dried and resuspended in 100 µL nuclease free water. Total RNA quality was assessed by a nanodrop measurement and an Agilent 2100 Bioanalyzer pico Chip.

### 3.4. rRNA Depletion

Ribosomal RNA was depleted with Ribo-Zero rRNA Removal Reagents (Bacteria)-Low Input and the Magnetic Core Kit-Low Input (Illumina, San Diego, California, United States) according to the manufacturer’s specifications, and purified using the Agencourt RNAClean XP Kit (Beckman Coulter Inc., Brea, California, United States). A pico chip run on the bioanalyzer 2100 confirmed the depletion of 16S and 23S rRNA.

### 3.5. Library Preparation and Sequencing

Library preparation was done with an Illumina TruSeq RNA stranded kit. All samples were pooled and sequenced on one lane of a Novaseq (paired-end, 150bp per read) at the Functional Genomics Centre Zürich. The data discussed in this publication have been deposited in NCBI’s Gene Expression Omnibus [81] and are accessible through GEO Series accession number GSE135966 (https://www.ncbi.nlm.nih.gov/geo/query/acc.cgi?acc=GSE135966)

### 3.6. RNAseq Analysis

All strains were previously sequenced in an unrelated project. Whole genome sequences are available under the NCBI Reference Sequence: NC_019556.1 for LL195, NZ_QYGN00000000.1 for stain N12-0605 and NZ_QYDR00000000.1 for stain N13-1507.

Sequenced RNA reads were aligned to their corresponding genomes using the BWA mem algorithm [82] with the default settings. Samtools [83] was used to convert and sort the output and featureCounts [84] summarized the aligned reads. Normalization and statistical calculation of differentially expressed genes was done in R with the EdegR package as described previously [85,86]. Differential expression (DE) was analyzed for each strain separately between the acid exposed sample and the control without acid. Genes or features with an FDR of <0.05 and a log2 fold change (logFC) of >1 or <−1 (representing a more than 2-fold up or down regulation) were considered significantly differentially expressed.

To identify regulatory elements a MEME search suing standard setting was performed [87,88] using the region 150-bp upstream of the start codons of differentially expressed genes. MEME motifs were filtered manually using the presence of a clear palindrome and the occurrence of a clear consensus sequence in multiple sequences as inclusion criteria. In addition, the ribosomal binding site was excluded. The subset was then subjected to a MAST analysis using the complete genomes of the 3 strains as input.

GO ontologies of genes were identified using eggNOG [89,90]. Relative enrichments were calculated using exact Fischer test available the topGO package in R [91]. Homologous proteins between the 3 strains were identified with a bi-directional best hit approach using the perl script “get_homologous” [92] and with standard settings.

### 3.7. Morphological Analysis

First visual inspection was done by microscopy of Gram stained cells. Acid stressed and control samples were fixed for 45 min using 1% (*v*/*v*) paraformaldehyde (PFA) in PBS directly after acid stress and stained according to standard protocols.

For flow cytometry, cells were also fixed using 1% (*v*/*v*) PFA in PBS, incubated for 45 min and DAPI (Sigma-Aldrich) was added to each sample at 1 µg/mL. The samples were then analyzed on an Image Stream X Mark II imaging flow cytometer (Merck Millipore, Burlington, Massachusetts, United States).

A tiered gating strategy was applied to gate for events in focus of the image stream and on single cells. These gates showed very accurate single cells in the control samples. However, a large quantity of aggregated cells were present in the samples after acid stress due to the intensive clumping. Gating these out would have removed more than 90% of the sample. These aggregates were therefore treated as a result rather than an artifact. Within the above-mentioned gates, Brightfield, darkfield and intensity of DAPI fluorescence was recorded for each cell in samples after acid stress and in controls. The Image Stream generated images of every event in flow which also enabled a manual validation of single events.

Motility assays were performed starting with liquid overnight cultures grown at 37 °C. A sterile 10 µL pipet tip was dipped into the culture and used to spot a small amount of culture onto 0.05% BHI soft agar plates. These were incubated for 24 h at 37 °C and photographed.

## Figures and Tables

**Figure 1 microorganisms-07-00455-f001:**
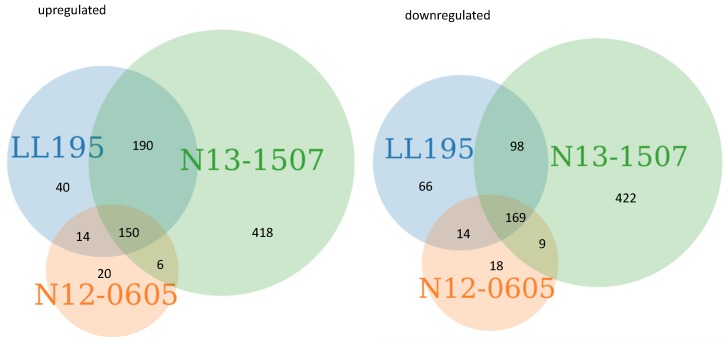
The number of genes that were up- and downregulated per strain in response to HCl acid shock at pH 3.

**Figure 2 microorganisms-07-00455-f002:**
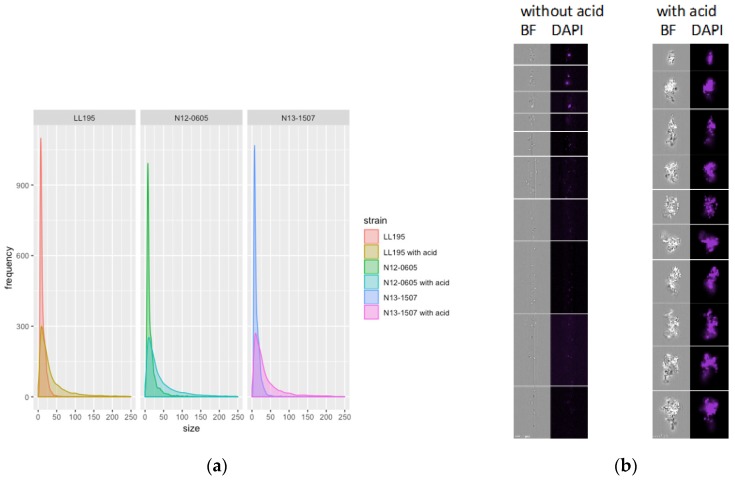
Morphological changes after HCl acid stress. (**a**) Flow cytometric analysis of cell size with and without acid. Each dataset represents 10’000 cells. (**b**) Image stream analysis of *L. monocytogenes* cells without (left) and with (right) acid stress. For each dataset, 10,000 cells were imaged. The figure contains representative images (left column: brightfield; right column: DAPI) from among the largest entities within each dataset.

**Figure 3 microorganisms-07-00455-f003:**
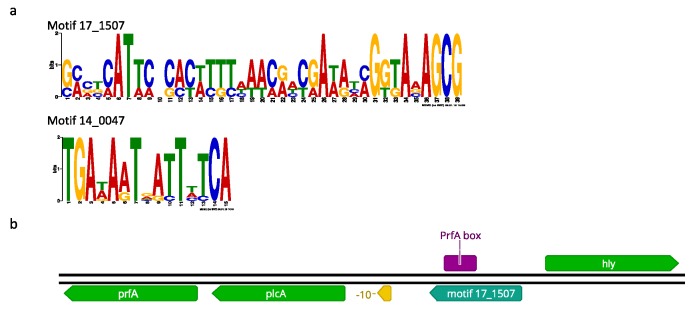
Motif search within 150 bp upstream of differentially expressed genes. (**a**) Sequence logos for motifs 17_1507 and 14_0047 (**b**) Schematic representation of the genomic context of motif 17_1507 in the promoter upstream of *plcA.* Coding sequences are not drawn to scale.

**Table 1 microorganisms-07-00455-t001:** Motifs identified within 150bp upstream of differentially expressed genes.

**Motif 17_1507**									
**Strain**	Motif sequence	Strand ^1^	e-value	Product up ^2^	Product down ^2^	FC up ^3^	FDR up ^4^	FC down ^3^	FDR down ^4^
LL195	GAAGAATATCCAAATGTGATTAAAAATATAGTTACACCG	reverse	9.16E-07	Cyclic-di-AMP phosphodiesterase GdpP	50S ribosomal protein L9	−0.220	0.774	−0.229	0.700
LL195	CGCTTTACCTGCTTCGGCGATTGAATGTCAGTATGAGGC	forward	2.19E-16	hypothetical protein	hypothetical protein	0.969	0.057	0.769	0.143
LL195	GAGGCATTAACATTTGTTAACGACGATAAAGGGACAGCG	reverse	1.81E-15	1-phosphatidylinositol phosphodiesterase	Listeriolysin O	0.802	0.107	3.013	0.000
LL195	CAAGAAATGGCGCTTTTCTTCAACAAAAAAGAGAAAGCA	reverse	1.18E-07	RNA polymerase sigma-H factor	Protein translocase subunit SecE	−0.650	0.227	0.679	0.281
LL195	GCAGAATACACCTATTTAGTAGGAGATAAAGTGAATGCT	reverse	1.48E-07	Internalin-A	hypothetical protein	0.079	0.904	0.283	0.686
LL195	CAAGAATTAGCAAATGTAAACGACGTAGCAATGGATGAG	reverse	6.03E-07	hypothetical protein	Nitrite transporter NirC	0.414	0.462	0.355	0.668
LL195	CACTTTAATGATATTAACGAAGAAGATGATTACTGGGGG	forward	3.67E-07	hypothetical protein	Glutathione reductase	1.576	0.001	1.218	0.028
LL195	ACAGAAATCAGCCACTTAATTAGCGAAACAATGACAGAG	reverse	7.55E-07	Putative TrmH family tRNA/rRNA methyltransferase	hypothetical protein	0.310	0.658	1.145	0.024
LL195	CTCTCTATCTTCAATGATTTTTGCAAGCGCGATTTGTTC	forward	8.88E-08	Glutathione amide reductase	Ribonuclease J 2	1.661	0.026	−0.411	0.591
LL195	GAGTCACTCACCCGCTTAAAAACAGATCACTGGAATGCG	reverse	1.65E-07	putative cyclic di-GMP phosphodiesterase PdeG	Malolactic enzyme	−0.527	0.348	−0.221	0.707
LL195	GCACCATTCGCACTCTTCAACATCGATATCGTTAGAGCG	reverse	2.93E-16	Ribonucleoside-diphosphate reductase 2 subunit alpha	hypothetical protein	1.620	0.013	0.141	0.842
LL195	CACTAATTCCCCCATTTATAAAATAAAATCGGTAAAGCG	reverse	2.78E-14	High-affinity heme uptake system protein IsdE	Iron-regulated surface determinant protein A	1.012	0.143	1.706	0.001
LL195	CGCTTGCTCTTTCTCATTGATTAATTGTTCATATTGCGC	forward	2.58E-07	putative protein YhaN	putative metallophosphoesterase YhaO	−0.614	0.282	−0.139	0.825
LL195	CCATAAGCGACATTATCATAAATCGAAAACGGGAATGGG	reverse	9.55E-07	Phosphate import ATP-binding protein PstB 3	Phosphate import ATP-binding protein PstB 3	0.767	0.360	0.468	0.447
LL195	CAGGCATTCAAACTTGCCAACAAATATACGAATAAAGCG	reverse	1.93E-07	Heptaprenyl diphosphate synthase component 2	3′,5′-cyclic adenosine monophosphate phosphodiesterase CpdA	−0.772	0.187	−1.617	0.005
N12-0605	GAACAAATCGCGCTTGCAAAAATCATTGAAGATAGAGAG	reverse	8.88E-08	Glutathione amide reductase	hypothetical protein	0.248	0.864	0.324	0.739
N12-0605	CTCTGTCATTGTTTCGCTAATTAAGTGGCTGATTTCTGT	forward	7.55E-07	Putative TrmH family tRNA/rRNA methyltransferase	Putative aminopeptidase YsdC	0.152	0.879	0.157	0.881
N12-0605	CCCCCAGTAATCATCTTCTTCGTTAATATCATTAAAGTG	reverse	3.67E-07	hypothetical protein	hypothetical protein	0.998	0.121	0.093	0.921
N12-0605	CTCATCCATTGCTACGTCGTTTACATTTGCTAATTCTTG	forward	6.03E-07	hypothetical protein	hypothetical protein	0.104	0.916	−0.116	0.940
N12-0605	AGCATTCACTTTATCTCCTACTAAATAGGTGTATTCTGC	forward	1.48E-07	Internalin-A	hypothetical protein	−0.204	0.855	−0.950	0.264
N12-0605	CGCTGTCCCTTTATCGTCGTTAACAAATGTTAATGCCTC	forward	1.81E-15	Listeriolysin O	1-phosphatidylinositol phosphodiesterase	2.764	0.012	0.729	0.267
N12-0605	GCCTCATACTGACATTCAATCGCCGAAGCAGGTAAAGCG	reverse	2.19E-16	hypothetical protein	hypothetical protein	0.639	0.420	0.753	0.289
N12-0605	CGGTGTAACTATATTTTTAATCACATTTGGATATTCTTC	forward	9.16E-07	Cyclic-di-AMP phosphodiesterase GdpP	Accessory gene regulator A	−0.356	0.721	−0.100	0.934
N12-0605	CGCTTTATTCGTATATTTGTTGGCAAGTTTGAATGCCTG	forward	1.93E-07	Heptaprenyl diphosphate synthase component 2	hypothetical protein	−0.704	0.298	−0.287	0.717
N12-0605	CGCTTTACCGATTTTATTTTATAAATGGGGGAATTAGTG	forward	2.78E-14	Iron-regulated surface determinant protein A	High-affinity heme uptake system protein IsdE	2.261	0.003	1.593	0.089
N12-0605	CGCTCTAACGATATCGATGTTGAAGAGTGCGAATGGTGC	forward	2.93E-16	Ribonucleoside-diphosphate reductase 2 subunit alpha	Ribonucleoside-diphosphate reductase subunit beta	1.063	0.211	0.957	0.204
N12-0605	CAAGAAATGGCGCTTTTCTTCAACAAAAAAGAGAAAGCA	reverse	1.18E-07	RNA polymerase sigma-H factor	Protein translocase subunit SecE	−1.214	0.027	0.495	0.603
N12-0605	GCTGAAGTAGCACTTGAAAAAGACGATATCGACTCTGCG	reverse	8.24E-07	Beta-barrel assembly-enhancing protease	hypothetical protein	0.029	0.977	1.196	0.081
N12-0605	CGCATTCCAGTGATCTGTTTTTAAGCGGGTGAGTGACTC	forward	1.65E-07	putative cyclic di-GMP phosphodiesterase PdeG	hypothetical protein	−0.688	0.296	−0.810	0.205
N12-0605	CCCATTCCCGTTTTCGATTTATGATAATGTCGCTTATGG	forward	9.55E-07	Phosphate import ATP-binding protein PstB 3	Phosphate-specific transport system accessory protein PhoU	−0.116	0.928	0.295	0.801
N13-1507	GAACAAATTGCGCTCGCAAAAATCATTGAAGATAGAGAG	reverse	8.60E-07	Glutathione amide reductase	hypothetical protein	4.418	0.000	2.365	0.000
N13-1507	CTCTGTCATTGTTTCGCTAATTAAGTGGCTGATTTCTGT	forward	7.55E-07	Putative TrmH family tRNA/rRNA methyltransferase	Putative aminopeptidase YsdC	0.314	0.290	0.011	0.981
N13-1507	CCCCCAGTAATCATCTTCTTCGTTAATATCATTAAAGTG	reverse	3.67E-07	hypothetical protein	hypothetical protein	1.047	0.005	1.084	0.001
N13-1507	CTCATCCATTGCTACGTCGTTTACATTTGCTAATTCTTG	forward	6.03E-07	hypothetical protein	hypothetical protein	−0.024	0.963	2.358	0.000
N13-1507	AGCATTCACTTTATCTCCTACTAAATAGGTGTATTCTGC	forward	1.48E-07	Internalin-A	hypothetical protein	0.482	0.353	NA	NA
N13-1507	CGCTGTCCCTTTATCGTCGTTAACAAATGTTAATGCCTC	forward	1.81E-15	Listeriolysin O	1-phosphatidylinositol phosphodiesterase	4.286	0.000	1.503	0.000
N13-1507	GCCTCATACTGACATTCAATCGCCGAAGCAGGTAAAGCG	reverse	2.19E-16	hypothetical protein	hypothetical protein	1.287	0.005	2.146	0.000
N13-1507	CGGTGTAACTATATTTTTAATCACATTTGGATATTCTTC	forward	9.16E-07	Cyclic-di-AMP phosphodiesterase GdpP	Accessory gene regulator A	−0.285	0.406	0.945	0.005
N13-1507	CGCTTTATTCGTATATTTGTTGGCAAGTTTGAATGCCTG	forward	1.93E-07	Heptaprenyl diphosphate synthase component 2	hypothetical protein	−1.672	0.000	−1.265	0.000
N13-1507	CGCTTTACCGATTTTATTTTATAAATGGGGGAATTAGTG	forward	2.78E-14	Iron-regulated surface determinant protein A	High-affinity heme uptake system protein IsdE	1.624	0.001	2.351	0.001
N13-1507	CGCTCTAACGATATCGATGTTGAAGAGTGCGAATGGTGC	forward	2.93E-16	Ribonucleoside-diphosphate reductase 2 subunit alpha	Ribonucleoside-diphosphate reductase subunit beta (thioredoxin two genes downstream)	1.508	0.000	1.653	0.000
N13-1507	TGCTTTCTCTTTTTTGTTGAAGAAAAGCGCCATTTCTTG	forward	1.18E-07	RNA polymerase sigma-H factor	putative protein YacP	−0.867	0.004	−0.786	0.026
N13-1507	GCTGAAGTAGCACTTGAAAAAGACGATATCGACTCTGCG	reverse	8.24E-07	Beta-barrel assembly-enhancing protease	hypothetical protein	−1.407	0.000	0.359	0.312
N13-1507	CGCATTCCAGTGATCTGTTTTTAAGCGGGTGAGTGACTC	forward	1.65E-07	putative cyclic di-GMP phosphodiesterase PdeG	hypothetical protein	−0.353	0.309	−0.288	0.421
N13-1507	CCATAAGCGACATTATCATAAATCGAAAACGGGAATGGG	reverse	9.55E-07	Phosphate import ATP-binding protein PstB 3	Phosphate import ATP-binding protein PstB 3	3.181	0.000	1.230	0.024
**Motif 14_0047**									
Strain	Motif sequence	Strand ^1^	Score	Product up ^2^	Product down ^2^	FC up ^3^	FDR up ^4^	FC down ^3^	FDR down ^4^
LL195	TGATAATAATTCTCA	reverse	4.17E-08	Iron(3+)-hydroxamate-binding protein YxeB	putative siderophore transport system ATP-binding protein YusV	0.222	0.820	0.531	0.341
LL195	AGAAAATCATTTTCA	forward	9.75E-07	Fe(2+) transporter FeoB	hypothetical protein	1.829	0.008	1.534	0.002
LL195	TGAGAATGATTTTCA	reverse	1.99E-08	Fe-S protein maturation auxiliary factor SufT	hypothetical protein	0.740	0.210	0.927	0.045
LL195	TGGTAGCCATTTTCA	forward	5.38E-07	hypothetical protein	putative protein YwqG	−1.006	0.048	−0.165	0.774
LL195	TGAAAACAATTTTCA	forward	3.78E-07	Phosphoglucomutase	Aldose 1-epimerase	−0.761	0.216	−0.624	0.289
LL195	TGGAAACAATTTTCA	forward	6.42E-07	hypothetical protein	hypothetical protein	0.893	0.209	0.812	0.239
LL195	TGAAAGTGATTTCCA	reverse	1.09E-07	O-acetyl-ADP-ribose deacetylase	Energy-dependent translational throttle protein EttA	2.519	0.000	−1.055	0.176
LL195	TGAAAATGATTTTCA	reverse	2.07E-09	Putative N-acetylmannosamine-6-phosphate 2-epimerase	Ribosomal RNA small subunit methyltransferase G	0.106	0.923	−0.749	0.137
N12-0605	TGAAAATCATTTTCA	forward	2.07E-09	PTS system glucitol/sorbitol-specific EIIA component	Fe(3+)-citrate-binding protein YfmC	0.103	0.927	1.426	0.012
N12-0605	TGAAAATTGTTTCCA	reverse	6.42E-07	hypothetical protein	Protein SapB	0.037	0.977	0.099	0.940
N12-0605	TGAAAATCATTCTCA	forward	1.99E-08	hypothetical protein	Fe-S protein maturation auxiliary factor SufT	0.710	0.281	1.343	0.037
N12-0605	TGAAAATTGTTTTCA	reverse	3.78E-07	Phosphoglucomutase	Nucleotide-binding protein YvcJ	−0.854	0.223	0.424	0.569
N12-0605	TGAAAATGGCTACCA	reverse	5.38E-07	hypothetical protein	Putative ion-transport protein YfeO	−0.542	0.500	−0.562	0.668
N13-1507	TGAAAATCATTTTCA	forward	2.07E-09	PTS system glucitol/sorbitol-specific EIIA component	Fe(3+)-citrate-binding protein YfmC	1.276	0.019	1.784	0.000
N13-1507	TGAAAATTGTTTCCA	reverse	6.42E-07	hypothetical protein	Protein SapB	3.061	0.000	3.743	0.000
N13-1507	TGAAAATCATTCTCA	forward	1.99E-08	hypothetical protein	Fe-S protein maturation auxiliary factor SufT	0.376	0.294	1.456	0.000
N13-1507	TGGTAGCCATTTTCA	forward	5.38E-07	hypothetical protein	putative protein YwqG	−0.605	0.178	0.357	0.364
N13-1507	TGAAAACAATTTTCA	forward	3.78E-07	Phosphoglucomutase	Aldose 1-epimerase	−1.563	0.000	−1.892	0.000

^1^ Orientation of the motif; ^2^ Gene product upstream/downstream of the motif; ^3^ Fold change of the gene upstream/downstream of the motif; ^4^ False discovery rate of the gene upstream/downstream of the motif.

**Table 2 microorganisms-07-00455-t002:** Strains used in this study.

Strain ID	Clonal Complex	Serotype	Isolation Source	Acid Phenotype	RNAseq Replicates
LL195	CC1	4b	Outbreak strain	Resistant	3
N12-0605	CC1	4b	Meat product	Resistant	3
N13-1507	CC6	4b	Blood	Resistant	2

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
