# Peer review of "Global Transcriptional Response of Three Highly Acid-Tolerant Field Strains of Listeria monocytogenes to HCl Stress"

_microorganisms, 2019, doi:10.3390/microorganisms7100455_

Round 1

Reviewer 1 Report

In this manuscript Horlbog et al describe an RNA-Seq analysis of 3 strains of L. monocytogenes (isolated from patients, or contaminated food) following acid treatment at pH 3 (using HCl). The authors identified differentially expressed genes (compared to non-acid treated control) and perform several informatic analyses on these genes. GO ontology identifiers suggest that the organisms undergoes remodeling that is consistent with growth stasis, elongation and acidification. The authors nicely rationalize how the observed gene expression patterns could correlate to the phenotypic changes in the bacteria following acid treatment. The authors then did a promoter analysis of acid-regulated genes and identified 2 potential regulatory sequence motifs that could be important for transcriptional responses to acid treatment.

In general, this was a well written and logically organized manuscript. Based on the transcriptional analyses a number of testable hypotheses were generated but not experimentally validated. Nevertheless, this manuscript represents an important contribution towards exploring mechanisms of acid tolerance. A paragraph stating the new hypotheses generated and suggestions for testing them would be welcome to reiterate that experimental validation is required.

Comments to authors:

The resolution on Figure 1 must be improved. It is not possible to read the axes or legends and the imagestream images at the top row of panel b appear empty. If DAPI was used to visualize, cells should be visible. If brightfield was used (cannot see if this is indicated) then both BF and DAPI images should be shown. Furthermore, a gating strategy that delineates single cells from elongated/aggregated cells should be implemented to allow quantification of  of events with and without acid treatment. Acidification of BHI medium to pH 1.6 was indicated to cause iron sequestration (lines 310-312). Would iron supplementation have affected the observed gene regulation? What other effects does acidification have on the medium? Could this influence metabolic gene expression? A legend is needed for the supplementary tables. Lines 242-243, additional justification is needed for the statement that downregulating ATP synthase transcripts increases the efficiency of using proton pumps to relieve acid stress.  Swarming assays were provided in supplemental figure 2. Could the same assay be used following acid treatment to test the predicted effects on motility?

Minor comments

Line 252 - "under organic acid stress", is HCl considered an organic acid? Lines 197-199 – the font size is reduced and "remarkably" is bolded. Has competition between wild type and constitutively active flaA been reported? How do the newly identified promoter motifs compare to the SigB binding site? Could SigB bind to these motifs?

Author Response

Dear reviewer 1,

Thank you very much for your thoughtful comments. They have been addressed point by point below:

The resolution on Figure 1 must be improved. It is not possible to read the axes or legends and the imagestream images at the top row of panel b appear empty. If DAPI was used to visualize, cells should be visible. If brightfield was used (cannot see if this is indicated) then both BF and DAPI images should be shown. Furthermore, a gating strategy that delineates single cells from elongated/aggregated cells should be implemented to allow quantification of  of events with and without acid treatment.

We think all the above is concerning figure 2, not 1. We have replaced the figure with one of better quality and added the corresponding DAPI images. The figure legend has been adjusted accordingly, and the gating strategy has been added to the materials and methods section.

Acidification of BHI medium to pH 1.6 was indicated to cause iron sequestration (lines 310-312). Would iron supplementation have affected the observed gene regulation? What other effects does acidification have on the medium? Could this influence metabolic gene expression?

To clarify, the BHI was adjusted to a final pH of 3.0 by adding BHI at pH 1.6 to the culture. We are currently working on follow-up experiments to this study. Your point about  supplementing iron is a good question that we will add to the list. would be an interesting addition. To our knowledge there are no other obvious effects of the pH on the medium.

A legend is needed for the supplementary tables.

We have added legends in separate sheets to both supplementary tables.

Lines 242-243, additional justification is needed for the statement that downregulating ATP synthase transcripts increases the efficiency of using proton pumps to relieve acid stress.

We have reformulated this into a clearer statement.

Swarming assays were provided in supplemental figure 2. Could the same assay be used following acid treatment to test the predicted effects on motility?

This is an experiment that we were also debating, however we assume that the effect of 1hr acid stress would be abolished after a few hours in soft agar at pH 7. Once the cells have adjusted back to pH 7.0 normal swarming behavior is expected and results would be very similar to the control condition.

We already tried to use soft agar at low pH and were we faced with technical problems due to the very liquid state of the low pH soft agar. It is also prone to producing a liquid layer on top which precludes any meaningful interpretation of the swarming behavior.

Minor comments

Line 252 - "under organic acid stress", is HCl considered an organic acid?

This has been corrected.

Lines 197-199 – the font size is reduced and "remarkably" is bolded.

Thank you for catching this, it has been corrected.

Has competition between wild type and constitutively active flaA been reported?

Not to our knowledge. The studies establishing MogR as a repressor of flagella genes (e.g.  https://journals.plos.org/plospathogens/article?id=10.1371/journal.ppat.0020030) show reduced virulence in mice in mogR knockout mutants, likely due to the overexpression of flaA in those strains. However, no competition experiments were performed, and the observed decline in virulence may not be exclusively due to flaA overexpression.

How do the newly identified promoter motifs compare to the SigB binding site? Could SigB bind to these motifs?

The newly identified motif has no similarities with the SigB consensus site (-35: GTTT, -10: GGGWAT, https://jb.asm.org/content/185/19/5722)

Reviewer 2 Report

Global transcriptional response of three highly acid-tolerant field strains of Listeria monocytogenes to HCl stress

Summary:

The authors investigate the acid stress response of three acid tolerant L. monocytogenes in early stationary phase by use of transcriptomic and morphological analysis. The data show a global transcriptional response resulting among others in growth arrest, reduced metabolism, the reduction of motility as well as upregulation of early-stage virulence genes. The observed formation of biofilm microcolonies indicates that cells are able to develop a more favorable microenvironment where they are able to survive acetic stress. Further, the authors suggest that the low pH could serve as a signal to induce the expression of early phase virulence genes during entry of the human stomach.

Broad comments:

Generally speaking, the manuscript is well-written, logically structured and contains relevant information in a concise manner. The study setup was carefully designed and overall obtained data support the conclusions of the manuscript. However, some corrections should be done:

Three L. monocytogenes isolates were investigated in this study including two closely related CC1 isolates as well as one CC6 strain. One major conclusion in the section 2.1 General response of 3 strains to acid stress was that the acid regulon of the CC6 strain N13-1507 was larger than that of the two CC1 strains (line 83-84). However, they do not discuss possible reasons for this observation including the likely relevance of the genetic background. The authors point out the considerably large core acid regulon consisting of 150 upregulated and 169 downregulated genes. In the results and discussion section they concentrate on specific genes in pathways that of special interest. However, it would be of added value to provide an overview on the number of differentially expressed genes grouped according to their main functions like metabolism or respiration, cell wall composition and motility or virulence genes. This information would help the reader to easily get a first impression on the global transcriptional response of the investigated isolates to acid stress. L. monocytogenes face acidic conditions during entry of the human stomach. Thus, it seems likely that acid tolerance provides certain advantages for the opportunistic pathogen to survive passage of the stomach. Indeed, findings in this study like upregulation of early stage virulence genes and downregulation of highly immunogetic flaggella support this hypothesis. However, care should be taken during discussion of this topic since tests were carried out under laboratory conditions and focused solely on investigation of acidic stress. Obtained data indicate but do not allow drawing definite conclusions on the behavior of L. monocytogenes during entry of the stomach like it has been done in the manuscript several times (e.g. lines 268-269, 283-285). Please provide information on the motility experiment in the Materials and Methods section.

Specific comments: 

Line 82: “:(i) a considerable proportion of 150 differentially expressed genes was shared across all strains”

In the lines above this statement the authors describe a core acid regulon consisting of 150 up- and 169 downregulated genes. Hence, the number of differentially expressed genes given in line 82 should be 319.

Lines 99-104: Genes should be formatted in italic stile.

Line 197: “Remarkably. The biofilm dispersion locus A…”

Please replace the dot by a comma.

Lines 208-209: “… while pre-exposure to acidic stress conditions did not the biofilm formation ability at neutral pH.”

Please change the sentence to: ““… while pre-exposure to acidic stress conditions did not affect the biofilm formation ability at neutral pH.”..

Line 268: “Combined with other studies, our data support that flagella are suppressed during…”

Please change to: “Combined with other studies, our data support the hypothesis that flagella are suppressed during…”.

Line 283-284: “Taken together, the exposure of L. monocytogenes to HCl during the gastric passage…”.

Please change to: “Taken together, our data indicate that the exposure of L. monocytogenes to HCl during the gastric passage…”.

Line 337: “… of the newly identified a motif…”

Please delete “a”.

Line 358: “… same batch of pre-prepared of BHI.”

Please delete the marked “of”.

Table 1: A title should be provided.

Figure 2a and b: The figures are of poor quality and in this stage not able to support the findings of the manuscript.

Author Response

Dear reviewer 2,

thank you very much for your comments. They are addressed point by point below:

Three L. monocytogenes isolates were investigated in this study including two closely related CC1 isolates as well as one CC6 strain. One major conclusion in the section 2.1 General response of 3 strains to acid stress was that the acid regulon of the CC6 strain N13-1507 was larger than that of the two CC1 strains (line 83-84). However, they do not discuss possible reasons for this observation including the likely relevance of the genetic background.

A sentence has been added to the paragraph to reflect this thought.

The authors point out the considerably large core acid regulon consisting of 150 upregulated and 169 downregulated genes. In the results and discussion section they concentrate on specific genes in pathways that of special interest. However, it would be of added value to provide an overview on the number of differentially expressed genes grouped according to their main functions like metabolism or respiration, cell wall composition and motility or virulence genes. This information would help the reader to easily get a first impression on the global transcriptional response of the investigated isolates to acid stress.

We were debating making the information in the overview sheets (“all up” / “all down”) in supplementary table 2 part of the main manuscript. Ultimately, we decided against it in order avoid adding bulk to the manuscript. Your comment is a good point, though, and we have added a reference to supplementary table 2 to this passage of the manuscript.

monocytogenes face acidic conditions during entry of the human stomach. Thus, it seems likely that acid tolerance provides certain advantages for the opportunistic pathogen to survive passage of the stomach. Indeed, findings in this study like upregulation of early stage virulence genes and downregulation of highly immunogetic flaggella support this hypothesis. However, care should be taken during discussion of this topic since tests were carried out under laboratory conditions and focused solely on investigation of acidic stress. Obtained data indicate but do not allow drawing definite conclusions on the behavior of L. monocytogenes during entry of the stomach like it has been done in the manuscript several times (e.g. lines 268-269, 283-285).

The tone of these hypotheses has been toned down across the manuscript.

Please provide information on the motility experiment in the Materials and Methods section.

A paragraph has been added to the morphological analysis section.

Specific comments: 

Line 82: “:(i) a considerable proportion of 150 differentially expressed genes was shared across all strains”

In the lines above this statement the authors describe a core acid regulon consisting of 150 up- and 169 downregulated genes. Hence, the number of differentially expressed genes given in line 82 should be 319.

This has been corrected.

Lines 99-104: Genes should be formatted in italic stile.

Thank you for catching this. We double-checked all gene and species names to make sure they are  formatted in italic.

Line 197: “Remarkably. The biofilm dispersion locus A…”

Please replace the dot by a comma.

This has been corrected.

Lines 208-209: “… while pre-exposure to acidic stress conditions did not the biofilm formation ability at neutral pH.”

Please change the sentence to: ““… while pre-exposure to acidic stress conditions did not affectthe biofilm formation ability at neutral pH.”..

This has been corrected.

Line 268: “Combined with other studies, our data support that flagella are suppressed during…”

Please change to: “Combined with other studies, our data support the hypothesis that flagella are suppressed during…”.

This has been changed.

Line 283-284: “Taken together, the exposure of L. monocytogenes to HCl during the gastric passage…”.

Please change to: “Taken together, our data indicate that the exposure of L. monocytogenes to HCl during the gastric passage…”.

We changed this sentence.

Line 337: “… of the newly identified a motif…”

Please delete “a”.

The “a” has been deleted.

Line 358: “… same batch of pre-prepared of BHI.”

Please delete the marked “of”.

This has been addressed.

Table 1: A title should be provided.

A title has been added to the table.

Figure 2a and b: The figures are of poor quality and in this stage not able to support the findings of the manuscript.

We have replaced the figure with better quality and added the corresponding DAPI images with a better description in the figure itself  and adjustments in the figure legend.